# Correlation between Milk Bacteriology, Cytology and Mammary Tissue Histology in Cows: Cure from the Pathogen or Recovery from the Inflammation

**DOI:** 10.3390/pathogens9050364

**Published:** 2020-05-09

**Authors:** Gabriel Leitner, Shlomo E. Blum, Oloeg Krifuks, Nir Edery, Uzi Merin

**Affiliations:** 1Emeritus Senior Scientist, National Mastitis Reference, Kimron Veterinary Institute, P.O.B. 12, Bet Dagan 50250, Israel; 2National Mastitis Reference Center, Department of Bacteriology, Kimron Veterinary Institute, P.O.B. 12, Bet Dagan 50250, Israel; shlomo.blum@mail.huji.ac.il (S.E.B.); krifol@walla.co.il (O.K.); 3Pathology, Kimron Veterinary Institute, Ministry of Agriculture & Rural Development, P.O.B. 12, Bet Dagan 50250, Israel; nire@moag.gov.il; 4Emritius Senior Scientist, Department of Food Quality and Safety, Institute of Postharvest and Food Sciences, The Volcani Center, P.O. Box 6, Bet Dagan 50250, Israel; uzmerin@gmail.com

**Keywords:** mastitis, cow, intermammary infection, PCR

## Abstract

The aim of the current study was to verify the existence of a significant correlation between bacterial isolation (or not) and mammary gland inflammation, using traditional bacterial culturing and PCR, milk leucocytes distributions, and tissue histology. Twenty-two cows were tested at the level of the individual gland for bacteriological culture and real-time PCR (RT-PCR), milk composition, somatic cells count (SCC), and cell differentiation. Post-slaughter samples of teat-ends and mammary tissues were tested for histology and bacteriology by RT-PCR. The 88 glands were assigned to either outcome: 1. Healthy—no inflammation and no bacterial finding (NBF) (n = 33); 2. Inflammation and NBF (n = 26); 3. Inflammation and intra-mammary infection (n = 22) with different bacteria. Bacteriology of milk samples and that of the RT-PCR showed 91.4% agreement. In the lobule’s tissues of healthy glands, ~50% were milk producers and the other glands had dry areas with increased fat globules with a low number of leukocytes. In contrast, ~75% of the infected glands were identified as inflamed, but with no isolation of bacteria. Infiltration of mononuclear cells and neutrophils into the connective tissue was observed but not in the lobule’s lumen. In summary, the study confirms that not every mastitis/inflammation is also an infection.

## 1. Introduction

In dairy cows, goats, and sheep, the importance of mastitis and its effects on animal health and welfare are well recognized. Mastitis leads to lower milk yield and quality, especially for industrial processing of dairy products, and thus it economically affects the whole dairy industry [1,2,3,4,5,6]. Two types of mastitis—inflammation of the mammary gland—can be distinguished: i) infectious, caused by intra-mammary infection (IMI), most often by bacterial pathogens; ii) non-infectious, associated mainly by milk stasis, normally at the end of the lactation, during estrus, after vaccination or other environmental or pathophysiological disturbances. The keys to this differentiation are the presence of an intra-mammary pathogen and leukocyte cell-type distribution, both often assessed by milk sampling. In infectious mastitis, pathogen identification to genus- and sometimes to species-level is highly important for short and long-term treatment, management, and prevention decisions in mastitis control. Treatment is obligatory to assure animal welfare in severe cases of clinical mastitis. Treatment often includes supportive care, non-steroidal anti-inflammatory drugs (NSAID) therapy to reduce inflammation, pain and fever, oxytocin therapy and frequent milk removal from the infected glands, and antibiotics therapy often used to prevent or contain the systemic spread of the pathogen. Most importantly, the use of antibiotics has become contradictory today, and even restricted in some countries. Antibiotic treatment in mastitis should be considered as a second line of treatment, justified only when bacteria are isolated and identified in affected glands. Often, bacteria are eliminated before mastitis is detected and/or the bacteria are resistant to available antibiotics [7]. Although clinical mastitis is usually clearly detected and defined, in many cases the animals are treated with antibiotics without proper diagnostic of the specific pathogen involved. Oliveira et al. [8] showed results stating that: “About, 35% of these treatments were given to cases which were culture negative at the time of detection and a further 17% were administered to cases for which there are no approved effective antimicrobials.”

Subclinical mastitis is defined as an inflammation of the mammary gland that causes changes in milk composition, increased somatic cell count (SCC), and alteration of leucocytes distribution. In most cases, but not necessarily always, subclinical mastitis is caused by an IMI. Subclinical mastitis is often unnoticed and more difficult to diagnose than clinical mastitis because no visual clinical symptoms exist and changes in milk appearance are less obvious if present at all. However, subclinical mastitis has the greatest economic consequences regarding reduced milk yield, lower milk quality [9,10], and the need for justifying antibiotic treatment, including at dry-off. In commercial dairy herds, subclinical mastitis refers to cows whose SCC increases to levels that influence bulk milk SCC and, therefore, may affect milk value for the farmer. Most of the cows with subclinical mastitis might be ignored if their milk is not expected to increase bulk tank SCC payment level cutoff. Only those cows diagnosed by the routine milking test with SCC > 1 × 10^6^ cells/mL usually need the attention of the farmer [11,12]. Consequently, most of the data and knowledge on subclinical mastitis as well as most of the experts’ suggestions regarding prevalence, bacteria type, influence on milk yield and quality, and treatment during lactation and at the dry-off period result from research findings. As a result of the different reference to subclinical mastitis, there is a gap between proposal and reality. For instance, a lot of the decisions on management practices regarding inflammation cases coming from the field rely on SCC or cow-side testing, where no bacteriology tests are performed. Thus, it refers to subclinical infection while it should have referred to it as subclinical inflammation, unlike the case in research. As a consequence, these assumptions lead to the incorrect decision of how to deal with subclinical inflammation.

The gold standard for evaluating the infection status of subclinical mammary glands is the isolation and identification of the same bacterial pathogen in at least two of three consecutive samplings [13]. Still, in many published studies, diagnosis relies on a single milk sample, usually taken by the treating veterinarian or the farmer, resulting in high proportions of false-positive interpretations, because bacterial growth is considered as an intra-mammary infection whereas it could also be due to sampling contamination or transient growth. This is often the case regarding selective dry-off treatment since only one sample is assessed on the last day of milking and two extra samples are taken only one- and two-weeks post-partum. Moreover, in many studies, cows with SCC < 200 × 10^3^ cell/mL are considered healthy (uninfected controls) and therefore are not tested for the presence of intra-mammary infections. On the other hand, it is acceptable that 20–30% of milk samples taken from clinical and subclinical infections result in no bacterial growth by standard culturing [14,15]. At the same time, multiple bacteria species (2–4), including major mastitis species in single samples from mastitis as well as animals presumed to be uninfected (low SCC) were detected as positive using polymerase chain reaction (PCR)[15], raising the question of possible sample contamination. Hence, the results of the dynamic nature of new infections, the overall prevalence of clinical and subclinical infections, and the effect of various treatments are wide but inconclusive. Thus, it raises questions like: 1. Is the definition of mastitis as mammary inflammation as distinct from intra-mammary infection clear? 2. How to distinguish milk sample contamination and teat-end contamination from intra-mammary infection? 3. Are modern laboratory methods and technologies used for the detection of intra-mammary infection not sensitive enough or overly sensitive? A study conducted with a small number of dairy heifers (96 glands) in their first lactation tested their bacteriological status, SCC and differential leucocyte count in milk, starting 60–120 days post parturition and every 50–60 days thereafter, until treated with antibiotics at drying-off [16]. Following the second partum, cows were tested again at 1 and 2 weeks and then monthly, up to 3 months. The major result of that study showed that “During lactation, 84.5% quarters had no change in their classification, 6.2% were newly infected with other pathogens, 3.1% were classified as self-cured and in 6.2% sporadic bacteria were isolated”. The same cows were involved in a large study that evaluated the efficiency of antibiotic treatment at dry-off. The bacteriological diagnosis was conducted in the same laboratory. Thus, two protocols compared cure and new infection rates: 1. One sample at drying-off and two samplings 1- and 2-weeks post-partum; 2. Three or more samplings during the lactation, then at 1 and 2 weeks, followed by monthly sampling up to 3 months post-partum. The results indicated significant differences between protocols 1 and 2: the percent of infected glands at dry-off (39.8% vs. 26.9%), cure post-partum (76.7% vs. 55.2%), and new infection (13.8% vs. 8.9%), respectively [17]. Due to these discrepancies, we changed our use of the definition of mastitis to inflammation and changes in leukocyte cell-type distribution with no bacteria isolation or with IMI with pathogen isolation in two consecutive samplings. Isolation of the same bacterial species, even in two samplings without inflammation, was not defined as a real mammary gland infection, i.e., it was regarded as sample contamination or teat-end bacteria. In order to answer questions 2 and 3 above, a study was conducted with dairy sheep and goats [18] comparing traditional bacterial culturing and real-time PCR (RT-PCR). The results clearly indicated that when using the suggested definition of mastitis, both bacterial culture and PCR methods proved to have 98.9% agreement for negative cultures (n = 30) and positive cultures (n = 30) (*Staphylococcus aureus,* coagulase-negative Staphylococcus (CNS), *Streptococcus agalactiae*, *Strep. dysgalactiae* and *Strep. uberis*). Other studies found lower agreement, probably due to contamination, which results in false-positive cultures for glands free of inflammation [15,19].

The aim of the current study was to verify the existence of a correlation between bacterial isolation (or not) and mammary gland inflammation, using traditional bacterial culturing and PCR, milk leucocytes distributions, and tissues’ histology at the slaughter of glands free of infection or infected with different bacteria.

## 2. Results

The average lactation, days in milk, and milk yield (L/d) of the 22 cows was 3.3 ± 1, 381 ± 32, 24.7 ± 1.4, respectively. According to the classification of inflammation (based on SCC level) and bacterial isolation (traditional bacterial culture) of the 88 glands, it was assigned to one out of three possible classes: 1. Healthy—no inflammation and no bacterial finding (NBF) (n = 33); 2. Inflammation and NBF (n = 26); 3. Inflammation and intra-mammary infection with CNS (n = 7), *Escherichia coli* (n = 13), Streptococci (n = 2). Seven glands were dried off (not milked and therefore not included). *E. coli* was isolated up to 15 days before the day of slaughter in all 13 glands, while only three glands were positive on slaughter day, and 10 of 13 samples negative on day of slaughter were considered as post-*E. coli* (PEC).

Mean and SE of SCC, milk composition, coagulation properties, and leucocytes distribution are summarized in Table 1. The milk of the healthy glands had SCC < 300 × 10^3^ cells/mL milk, ~50% leucocytes, >76% casein, and high CF. In comparison, cows diagnosed with inflammation but with no pathogen (class 2) had SCC > 1 × 10^6^ cells/mL milk, ~80% leucocytes, ~72% casein and the milk of most of them did not coagulate. All the infected glands (class 3) had increased SCC, 70–80% leucocytes, and 71–73% casein. CF was moderate in CNS-infected glands and very low to no coagulation was observed in *E. coli* and Streptococci-infected glands.

Comparing the results of the traditional bacteriology in milk and that of RT-PCR showed agreement between tests in 91.4% of the glands (74/81). Two out of 7 CNS, 2 out of 3 *E. coli,* and 1 out of 2 Streptococci-positive glands in classical bacteriology were negative by the RT-PCR. In NBF glands, 2/69 (59 + 10 PEC) were negative in the classical bacteriology and positive by RT-PCR (Table 2). The results of the RT-PCR of the tissue samples showed that from 7 CNS-positive samples in culture, 6 were positive for CNS by RT-PCR: 4 at the teat-end and in the lobules, 1 at the teat-end alone, and 1 in the lobules alone. Of the three *E. coli* glands that were positive at the day of slaughter in culture, 2 were also positive for *E. coli* at the lobules. Of the 10 PEC, all were negative for *E. coli* in RE-PCR, while 1 was positive for CNS at the teat-end and 1 for Streptococci at both tissue sites. The two Streptococci-culture-positive glands were found positive for CNS by RT-PCR. Of the 59 NBF glands, 16 tissue samples were CNS-positive by RT-PCR, all in only one of the tissue sites, and most of those were assigned as inflamed glands (class 2, 9/26; 34.5%).

Microscope photographs (Figure 1 and Table 3) revealed no abnormal tissues and increased inflammation at the teat-end of most of the glands regardless of inflammation and isolation status (Figure 1A). In the lobule’s tissues of the healthy glands (class 1), ~ 50% were milk producers (Figure 1B) and the other glands had dry areas with increased fat globules (Figure 1C) with only a low number of leukocytes. In contrast, in ~75% of the glands identified as inflammation but with no bacteria isolation (class 2), infiltration of mononuclear cells and neutrophils into the connective tissue was observed but not in the lobules lumen (Figure 1D, 1E). In addition, in a low number of glands, (6/26), a proliferation of blood vessels was observed at the teat-end. Of the infected glands (class 3), ~50% were milk producers and the other glands had dry areas with an increased number of fat globules. Regardless of milk production condition, in ~75% of the infected glands infiltration of mononuclear cells and neutrophils into the connective tissues, but not in the lobules was observed. Only three glands, 2 infected with *E. coli* and 1 with CNS showed a high number of neutrophils in the lobules (Figure 1F).

## 3. Discussion

The aim of this study was to assess the correlation between intra-mammary infection detection by classical bacteriological culture and RT-PCR to mammary gland inflammation. This work intended to raise awareness of three critical issues in mastitis diagnosis, management, and research: the definition of mastitis as mammary inflammation in contrast to intra-mammary infection, milk sample collection protocol, and use of laboratory methods and technologies. Modern developments in detection methods of mastitis and IMI (e.g., PCR) and control and the abundant research studies and overall interest in mastitis, prompt for a general agreement on clear definitions of mastitis, IMI, and udder health status between dairy farmers, veterinarians, the dairy industry, and researchers. Mastitis refers to inflammation of the mammary gland, whereas IMI refers to the presence of an infective pathogen in the mammary gland [13]. “True” IMI will lead to mastitis and mastitis in most cases results from IMI. However, because diagnostic and treatment methods have different targets, mechanisms, rates of success, and consequences, it is important to make the distinction between these two processes clear and generally accepted in all scientific and professional jargon and literature. Inflammation of the mammary gland indicates a certain disturbance in the mammary gland function and homeostasis, which results in negative changes in milk composition, such as decreased lactose level, increased ions concentration, impaired coagulation properties, and increased SCC and altered cell types distribution in comparison to a healthy gland [5,20,21]. Most of the damage in terms of milk quality, properties, value and yield results from the inflammatory process, which in certain situations may be present even in the absence of a current IMI [22]. Therefore, the inflammation should be the major target for recovery or healing. The decision on how to manage the inflammation, either to treat or to “ignore” it, must consider animal welfare, potential hazards for human milk consumption (e.g., zoonotic pathogens, severe milk alterations), and the economic impact. If animal welfare and milk safety are not compromised and treatment is not economically justified, then the inflammation could be accepted for the time being. This is, in fact, the common practice, and most subclinical mastitis cases are routinely ignored, even though they may negatively affect milk yield and reproduction efficiency at the animal level [23,24] and decrease milk quality for dairy industrial processing [20,21]. New regulations for antibiotic use in dairy farms restrict the ability to treat IMI during lactation and subclinical mastitis during the dry-off period. Many studies use the 200 × 10^3^ cells/mL of milk as the “universal” SCC cutoff value to define “healthy” milk from individual glands, cows, and bulk milk tanks. It is clear, however, that the same cutoff cannot equally apply at these latter three levels. The bulk milk level is the problematic one due to mixing all of the cows, milk into it, meaning that the SCC value is actually only a very rough estimate of the average SCC of all the milked glands. In the current study, for instance, SCC was up to 300 × 10^3^ cells/mL in healthy glands, with no bacteria found in milk by either classic bacteriology or PCR (i.e., no IMI), normal leucocytes distribution, normal % casein and high CF values in the glands’ milk, and normal histology observed in the lobule’s tissues. The understanding when and how to consider mastitis as a disease that needs immediate attention should rely on more than SCC and must be adjusted to the source of the milk: gland, cow, or bulk tank.

Isolation of the pathogen that initiated the inflammation response is highly important for management decisions at the herd level and deciding on the individual cow’s treatment. Due to the natural behavior of the pathogen involved (e.g., intermittent secretion in milk), the pathophysiology of the response and the accepted routine method for milk sampling via the teat canal, which is often colonized or contaminated by potentially mastitis pathogens, many samples result in contaminated, false-positive or false-negative cultures. For this reason, it is advisable that two out of three consecutive samplings should be positive for the same pathogen to confirm the infection status of a mammary gland [13], but even this should be carefully interpreted if there are no indicators of an inflammatory response in the gland. This subject is probably the cause of the large differences regarding the prevalence of IMI, unjustified treatments, and above all, reported cure rates. Cure of IMI is related to the detection of a pathogen in relation to time, i.e., isolating at “time zero” and not afterward. However, in many studies, only one sampling is done at “time zero” and even when performed more than once, sampling is repeated after days or weeks. In relation to bacteria clearance alone, the term recurrence of IMI, implying cure between infections, might not apply for infection of the same gland and by the same pathogen weeks apart. This issue is highly important when using antibiotics during lactation or at drying-off. All in all, bacteria cure is undoubtedly important for the final healing process. In the current study, a high number of glands clearly with ongoing inflammation (>1 × 10^6^ cells/mL milk, ~80% leucocytes, ~72% casein, and lack of milk coagulation) were actually free of pathogens as determined be classic bacteriology and PCR. In the 10 glands from which *E. coli* was isolated, no bacteria were found in milk and tissues at the time of slaughter. The history of infections in most of these glands is unknown besides previous *E. coli* infection, but the inflammation found after the slaughter was most probably related to ‘post-infection’ because a similar process was observed in different cows with healthy glands. Interestingly, the lobule’s tissues were similar to those found in the healthy glands with an association to the stage in lactation, except for the PEC glands, in which infiltration of mononuclear cells and neutrophils in the connective tissue but not in the lobules was observed. These results suggest that in many cases, bacteria clearance is only the first stage of a long course of healing. Yet, because of the low coagulation properties, milk of such glands is of lower value for the cheese industry, and if milked into the bulk tank, regardless of the bulk tank SCC, it lowers the whole tank’s milk quality for industrial processing [25]. In the current study, according to the three consecutive weekly samples of the infected glands, high agreement existed between classic bacteriology and PCR in milk and in PCR of the tissues. The increased levels of SCC, with 70–80% leucocytes, lower % casein of the total protein (71–73%), and moderate CF for CNS infection and very low to no coagulation for *E. coli* and Streptococci, reinforces previous findings [5,26]. Here, in most of the infected glands, mononuclear cells and neutrophils infiltration in the connective tissue but not in the lobules was observed. Results confirmed that bacterial findings were real IMI associated with inflammation.

The question regarding milk sample collection and laboratory methods and technologies used also calls for attention. The classic bacteriology for detecting bacteria in milk are calibrated with the knowledge that because milk sampling is through the teat-canal and in a non-sterile environment, contamination might occur no matter how aseptic the procedure is. The contamination can occur from the cow’s skin and the sampler’s hand, and due to natural bacterial colonization or contamination in the teat-end. It is generally accepted to use only 0.01 mL of milk in bacterial culture (i.e., one regular bacterial loop); therefore, each colony represents 100 bacteria CFU/mL in milk. For most mastitis pathogens, the growth of a few colonies (some thousands of CFU/mL) is required in order to be considered significant growth and indicate this pathogen as the most probable infecting one in the tested gland. Best practice dictates that interpretation of results and the eventual outcome in terms of treatment and management decisions depend on the type of bacteria and the number of consecutive positive samplings, i.e., 2–3 times. The use of DNA-based methods, such as RT-PCR has increased the sensitivity and shortened the time for bacterial identification. However, results should be interpreted carefully because a positive PCR result does not necessarily mean that an active infection exists (i.e., viable bacteria), and it suffers from the same problems of contamination, but on an even higher magnitude. Therefore, results in prevalence and cure studies relying on PCR methods alone must be carefully and critically interpreted [15,18,19], and be even stricter depending on the number of repeated sampling and presence or not of inflammation indicators tested. In the current study, the overall agreement of the three repeated milk samplings between conventional bacteriological culture and PCR was over 90%. However, PCR performed on mammary tissues, resulted in some CNS positive in healthy glands and glands with inflammation and NBF, which can be contamination or can explain the cause of inflammation and the existence of glands with NBF. Preferably, PCR based diagnostics of IMI should aim at the detection of pathogens that require a short turn-over time for decision making (e.g., highly infectious or zoonotic pathogens), for screening purposes. Therefore, PCR should be calibrated with culture results through quantitative data, and coupled to evidence of inflammation during diagnosis. The same requirements of three repetitive testings might apply depending on the pathogen found. Tests resulting in multiple species should not be relied on. In addition, in the case of environmental pathogens, PCR targets should be as specific as possible to markers differentiating mammary-pathogenic subpopulations from the general population, when this knowledge is available [27].

## 4. Materials and Methods

This study was approved by the Institutional Animal Care Committee of the Agricultural Research Organization, The Volcani Center, Bet Dagan (Permit no. 59315).

### 4.1. Animals and Study Layout

Twenty-two Israeli Holstein multiparous cows at the Agricultural Research Organization, Volcani Center dairy herd entered the study. All cows were more than 200 days in lactation, not pregnant, and producing ~20–30 L milk/d. Cows were milked thrice daily (05:00, 13:00 and 20:00) in a milking parlor equipped with an on-line computerized AfiFarm Herd Management data acquisition system including AfiLab milk analyzer (Afimilk, Afikim, Israel; http://www.afimilk.com). The average milk yield of the herd was >11,500 L over 305 days. Food was offered ad-lib in mangers located in the sheds. Cows assigned by the farm manager for culling were tested by glands at 3 weekly consecutive samplings for bacteriology, milk composition, SCC, and cell differentiation. Cows were transferred to a slaughterhouse and sampled post-slaughter. The teat area was cleaned and disinfected with 70% alcohol, then the teat-ends and up to 10 cm of the mammary tissue dorsal to the teat connection to the gland were removed sterilely, stored individually in sterile bags and kept on ice until processing. The glands were classified as: 1. Healthy—no inflammation (SCC < 3 × 10^5^, PMN < 50% and macrophages < 20%, lactose ~50 g/L, curd firmness > 8.9) and NBF; 2. Inflammation and NBF; 3. Inflammation and intra-mammary infection.

### 4.2. Sample Collection and Analyses

#### 4.2.1. Milk

For bacteriological tests, the teats were cleaned and disinfected, the first milk streaks discarded and 3-mL samples of foremilk were taken. For other tests, separate glands were milked into containers, the milk volume was recorded, gently mixed and 0.5–1.0 L was taken for further analyses: SCC with the Fossomatic 360 (Foss Electric, Hillerød, Denmark) and gross milk composition, i.e., protein, casein, fat and lactose contents, with the Milkoscan FT6000 (Foss Electric). Analyses were performed at the Israel Cattle Breeders Association Laboratory (Caesarea, Israel). Leukocytes differentiation was performed by flow cytometry (FACs Calibur, Becton-Dickinson, San Jose, CA, USA) as described [28]. Rennet clotting time (RCT; min) and curd firmness (CF; V) after 60 min were tested using the Optigraph (Ysebaert, Frepillon, France) as described [29].

#### 4.2.2. Tissues

Upon arrival at the laboratory (~6 h), each gland sample was washed three times with sterile distilled PBS and the skin was disinfected with 70% alcohol and left to dry. Two sections were cut: the edge of the teat and gland lobules. From each section, 2 slices were cut: 1. For histological analysis, which was fixed in neutral buffered 4% formaldehyde of embroidery, and 2. For bacteria PCR.

### 4.3. Bacteriology, Histology and PCR Analysis

Bacterial culture and identification from milk samples were conducted according to the International Dairy Federation [30]. Briefly, 10 µL of milk was streaked on blood agar (nutrient agar with 5% washed sheep red cells) and McConkey plates and incubated for up to 48 h aerobically. Milk bacterial DNA was extracted using the Milk Bacterial DNA Isolation Kit (Norgen Biotek Corp., Thorold ON, Canada) with 350 μL of milk as the starting volume and included enzymatic pre-lysis and lysis steps to disrupt the cell walls of gram-positive and gram-negative bacteria, as well as spin column-based DNA purification and elution steps. Total DNA was extracted from tissue samples with the QIAGEN DNeasy Blood & Tissue Kit (QIAGEN, Hilden, Germany) following pre-lysis treatment for gram-positive bacterial cell disruption and complete enzymatic lysis prior to DNA extraction and purification. Direct bacterial DNA identification from milk and tissue samples DNA was performed with a commercial RT-PCR kit (PathoProof Mastitis Complete-12 kit PCR Assay; Thermo Fisher Scientific, Vantaa, Finland). The assay was carried out following the manufacturer’s procedure in a CFX96 Touch Real-Time PCR Detection System (Bio-rad, Rishon Le Zion, Israel). For interpretation of results, the following thresholds were used: results with Ct ≤ 33 were considered positive, results with Ct 34–36 were considered “weak” and results with Ct ≥ 37 were considered negative.

For histological analysis, sections were cut (4 μm) and stained with hematoxylin and eosin (H&E). Slides were viewed using a light microscope (Nikon Eclipse E600, Melville, NY, USA) and images were captured using the MagnaFire Digital Camera (Optronics, Goleta, CA, USA) controlled by NIS-Elements F3.0 software (Nikon, Japan).

### 4.4. Statistical Analysis

Statistical analyses were carried out with JMP software (SAS Institute Inc., Cary, NC, USA). The analyses performed were on the gland level. The effects of the bacteria type (gland classes 1–3; fixed effects) on the analyzed parameters were determined by ANOVA in a random design. The analyzed parameters were the fat (g/L), protein (g/L), lactose (g/L), SCC × 10^3^, leucocyte differentiation (%), RCT, CF.

The statistical model was:Y_ijk_ = μ + α_i_ + B_j_ + D_k_[B_j_] + e_ijk_
where: μ = mean of all data, α_i_ = the difference between the bacteria specie i from the trial mean, B_j_ = variance between cows, D_k_[B_j_] = variance between dates within a cow, e_ijk_ = residual variance between measurements (random error).

## 5. Conclusions

Mastitis remains and will probably remain an intrinsic part of dairy production. In order to refine management, prevention, and treatment measures, and define generally accepted guidelines for the possible hazards to human health, animal welfare, and economic effects of mastitis, some issues should be revised. 1. What is the level of inflammation on a gland/cow that should not be approved for human consumption; 2. What are the acceptable levels of inflammation that does not compromise animal welfare, milk quality and does not economically affect the dairy farm and industry; 3. What pathogens are potential human hazards and/or potentially of higher risk of infectivity in the heard (at subspecies level when appropriate). Responses to these questions can focus on the key issues, which milk is not being milked into the bulk tank regarding the level of inflammation and the contribution of SCC into the bulk milk, and which of the bacteria and/or level of inflammation has a low economical influence and can be accepted for the time being as part of the basic health.

## Figures and Tables

**Figure 1 pathogens-09-00364-f001:**
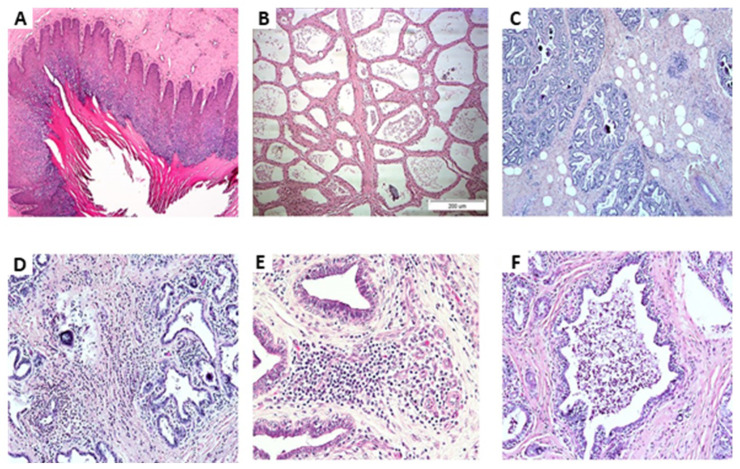
Microscope photographs: (**A**) Teat-end; (**B**) Lobule’s tissues of healthy glands of milk-producing cows; (**C**) Lobule’s tissues of healthy glands with non-milk producing areas and increased fat globules, with only a low number of leukocytes; (**D**,**E**) Lobule’s tissues of glands identified as inflamed but with no bacteria isolation, infiltration of mononuclear cells and neutrophils into the connective tissue, but not in the lobules lumen; (**F**) lobule’s tissues with high numbers of neutrophils.

**Table 1 pathogens-09-00364-t001:** Mean and SE of somatic cell count (SCC), milk composition, coagulation properties, and leucocytes distribution in the milk of 81 glands of 22 cows according to bacteria type (no inflammation and no bacterial funding (NBF), inflammation and NBF, inflammation and isolation of coagulase-negative staphylococci (CNS), Streptococci (Strep.) or *E. coli*.

Variable	No Inflammation and NBF (n = 33)	Inflammation and NBF (n = 26)	Inflammation and Isolation of CNS (n = 7)	Inflammation and Isolation of *E. coli* (n = 13)	Inflammation and Isolation of Strep. (n = 2)
SCC (× 10^3^)	185 ± 34 ^a^	1152 ± 140 ^b^	656 ± 175 ^ab^	1440 ± 677 ^b^	600 ± 2 ^ab^
CD18^+^ (%)	48.9 ± 3.7 ^a^	78.5 ± 2.5 ^b^	74.9 ± 4.5 ^b^	78.3 ± 2.5 ^b^	72.0 ± 8.7 ^b^
PMN (%)	25.9 ± 2.9	34.4 ± 4.3	36.9 ± 10.5	38.1 ± 6.1	15.5 ± 9.2
CD14^+^ (%)	14.3 ± 2.2	29.9 ± 4.1	18.3 ± 7.5	17.7 ± 4.0	30.0 ± 10.9
CD4^+^ (%)	3.1 ± 0.5	4.6 ± 0.8	4.4 ± 1.0	7.2 ± 1.2	5.0 ± 1.1
CD8^+^ (%)	4.9 ± 0.9	4.6 ± 1.2	5.2 ± 1.4	7.4 ± 1.3	3.5 ± 0.5
Fat (g/L)	33.9 ± 2.9	37.7 ± 2.7	36.3 ± 5.2	35.3 ± 3.3	46.9 ± 6.5
Protein (g/L)	40.0 ± 0.9	36.2 ± 1.1	38.3 ± 1.8	39.6 ± 0.9	30.4 ± 4.4
% casein	76.2 ± 0.3	71.9 ± 1.6	73.8 ± 1.0	73.8 ± 0.4	72.1 ± 2.5
Lactose (g/L)	47.7 ± 0.8 ^a^	33.2 ± 2.4 ^c^	43.4 ± 2.8 ^ab^	44.0 ± 1.0 ^ab^	38.7 ± 4.2 ^bc^
RCT (sec)	2429 ± 218 ^b^	4723 ± 156 ^a^	3157 ± 683 ^b^	4414 ± 299 ^a^	>5000 ^a^
CF (V)	9.70 ± 1.0 ^a^	0.77 ± 0.27 ^c^	6.48 ± 2.44 ^ab^	2.38 ± 0.93 ^bc^	0 ^c^

^a^^–c^ Means within rows with no common superscript differ significantly (*p* < 0.05).

**Table 2 pathogens-09-00364-t002:** Comparison of traditional bacteriology and RT-PCR in the milk and tissues of 81 glands of 22 cows according to bacteria type (no inflammation and no bacterial funding (NBF), inflammation and NBF, inflammation, and isolation of coagulase-negative staphylococci (CNS), Streptococci (Strep.) or *E. coli* or post-*E. coli.*

	No.	Milk		Tissues	
Gland’s class		Bacteriology	RT-CR	Edge of Nipple	Lobules
No inflammation and NBF	33	0	1 (Strep.)	4 (3-Strep., 1-CNS)	3 (CNS)
Inflammation	26	0	1 (*E. coli*)	9 (CNS)	4 (CNS)
Inflammation and isolation of CNS	7	7	5 (CNS)	5 (CNS)	5 (CNS)
Inflammation and isolation of *E. coli*	3	3	1 (*E. coli*)	1 (*E. coli*)	2 (*E. coli*)
Inflammation and post isolation of *E. coli*	10	0	0	1 (Strep.)	1 (CNS)
Inflammation and isolation of Strep.	2	2	1 (Strep.)	1 (CNS)	2 (CNS)

**Table 3 pathogens-09-00364-t003:** Microscope photographs of tissue sections of 81 glands of 22 cows according to bacteria type (no inflammation and no bacterial funding (NBF), inflammation and NBF, inflammation, and isolation of coagulase-negative staphylococci (CNS), Strep. or *E. coli* or post-*E. coli.*

Gland’s Class	No.	Edge of the Nipple	Lobules
No inflammation and NBF	33	Normal	50% normal production 50% connective tissue with increased fat globules
Inflammation	26	75% Normal 25% Proliferation of blood vessels	25% normal 75% Infiltrations of mononuclear cells and neutrophils in the connective tissue were observed but not in the lobules
Inflammation and isolation of CNS	7	Normal	75% glands were identified with infiltration of mononuclear cells and neutrophils in the connective tissue but not in the lobules
Inflammation and isolation of *E. coli*	3	Normal	High number of neutrophils in the lobules
Inflammation and post-isolation of *E. coli*	10	Normal	Glands were identified with infiltration of mononuclear cells and neutrophils in the connective tissue but not in the lobules
Inflammation and isolation of Strep.	2	Normal	glands were identified with infiltration of mononuclear cells and neutrophils in the connective tissue but not in the lobules

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
