# Peer review of "Correlation between Milk Bacteriology, Cytology and Mammary Tissue Histology in Cows: Cure from the Pathogen or Recovery from the Inflammation"

_pathogens, 2020, doi:10.3390/pathogens9050364_

Round 1

Reviewer 1 Report

The current research paper as written is interesting, and would be valuable contribution for scientific community dealing with mastitis, very common problem in dairy industry and milk production worldwide. This paper dealing with verifing the existence of a significant correlation between bacterial isolation (or not) and mammary gland inflammation, using traditional bacterial culturing and PCR, milk leucocytes distributions and tissues histology. The title of the paper and the purpose are properly defined. Generally, the study is well-designed and performed with adequate methodology, especially statistics, and results could be of a practical value. The results section is correctly written and consequently, the conclusions are adequate and of relevance for scientific and practical communities, particularly in dairy cattle management. The use of English is mostly very good. In general, the study design and analyses appear to be sound. Therefore, the undertaken issue is interesting and worth of disseminating and clarity of obtained results which could be of importance to the field of dairy science and production. However, from 30 used references the authors used only 8 in the last 5 years (from the year 2015). So, I suggest that the especially Introduction and Discussion sections improve with several more recent references. My suggestions are following:

Microbiological monitoring of mastitis pathogens in the control of udder health in dairy cows. Slov. Vet. Res. 53, 131-140.

The role of oxidative stress and inflammatory response in the pathogenesis of mastitis in dairy cows. Mljekarstvo 67, 91-101. doi.org/10.15567/mljekarstvo.2017.0201

Bovine mastitis: a persistent and evolving problem requiring novel approaches for its control - a review  Vet. arhiv 88, 535-557.  DOI: 10.24099/vet.arhiv.0116

Paraoxonase 1 in bovine milk and blood as marker of subclinical mastitis caused by Staphylococcus aureus. Res. Vet. Sci. 125, 323-332. doi: 10.1016/j.rvsc.2019.07.016

 In my opinion the current manuscript deserves by any means to be accepted for publication following acceptance of mentioned suggestions.

Author Response

Reviewer 1

We read all the articles suggested by the reviewer for inclusion in the manuscript. Unfortunately, the scope of these articles does not fit within the scope of our manuscript. Thus, none of it was included in the revised version.

Author Response

Reviewer 2.

No comments were raised.

Round 2

Reviewer 2 Report

Thank you for the changes. However, the central problem with the paper does not seem to be solved. Without making it clear how you have carried out the categorization precisely and providing references to the criteria - especially the thresholds used - the paper can not be evaluated conclusively.

l. 36 This is an assumption. You cannot say "cause". Either quote a passage from literature or speak of "association with it".

l. 73 I don't understand what you mean by the new sentence.

l. 148 Table 1: 1st column: mean SCC of 185,000 /mL in the group you classified as "no inflammation". This SCC is a strong indicator for inflammation.

l. 318 A clear definition is still missing. What means (low SCC (threshold), and standard cell distribution (PMN vs. MAC vs. LYM), coagulation properties (CT threshold etc.) and lactose level (%))

Author Response

Comments and Suggestions for Authors

Thank you for the changes. However, the central problem with the paper does not seem to be solved. Without making it clear how you have carried out the categorization precisely and providing references to the criteria - especially the thresholds used - the paper can not be evaluated conclusively.

  1. 36 This is an assumption. You cannot say "cause". Either quote a passage from literature or speak of "association with it".

AU: You are correct regarding non-infection and we changed it as suggested. However, in the case of bacteria isolation - the bacteria are the cause of the infection.

  1. 73 I don't understand what you mean by the new sentence.

AU: We tried to explain it better as follows: “As a result of the different reference to subclinical mastitis, there is a gap between proposal and reality. For instance, a lot of the decisions on management practices regarding inflammation cases coming from the field rely on SCC or cow-side testing, where no bacteriology tests are performed. Thus, it refers to subclinical infection while it should have referred to it as subclinical inflammation, unlike the case in research. As a consequence, these assumption lead to incorrect decision of how to deal with subclinical inflammation”.

  1. 148 Table 1: 1st column: mean SCC of 185,000 /mL in the group you classified as "no inflammation". This SCC is a strong indicator for inflammation.
  2. 318 A clear definition is still missing. What means (low SCC (threshold), and standard cell distribution (PMN vs. MAC vs. LYM), coagulation properties (CT threshold etc.) and lactose level (%))

AU: We added values to our definition of healthy glands.